# Recent 1D and 2D TD–NMR Pulse Sequences for Plant Science

**DOI:** 10.3390/plants10050833

**Published:** 2021-04-21

**Authors:** Tatiana Monaretto, Tiago Bueno Moraes, Luiz Alberto Colnago

**Affiliations:** 1Instituto de Química de São Carlos, Universidade de São Paulo, Av. Trabalhador São-Carlense, 400, São Carlos 13566-590, SP, Brazil; tatiana.monaretto@gmail.com; 2Embrapa Instrumentação, Rua XV de Novembro, 1452, São Carlos 13560-970, SP, Brazil; 3Departamento de Química, Instituto de Ciências Exatas, Universidade Federal de Minas Gerais, Av. Antônio Carlos, 6627, Belo Horizonte 31270-901, MG, Brazil; tiago.moraes@qui.ufmg.br

**Keywords:** time domain NMR, Carr-Purcell-Meiboom-Gill (CPMG), CWFP, relaxation measurement: pulse sequence

## Abstract

Time domain nuclear magnetic resonance (TD–NMR) has been widely applied in plant science in the last four decades. Several TD–NMR instruments and methods have been developed for laboratory, green-house, and field studies. This mini-review focuses on the recent TD–NMR pulse sequences applied in plant science. One of the sequences measures the transverse relaxation time (T_2_) with minimal sample heating, using a lower refocusing flip angle and consequently lower specific absorption rate than that of conventional CPMG. Other sequences are based on a continuous wave free precession (CWFP) regime used to enhance the signal-to-noise ratio, to measure longitudinal (T_1_) and transverse relaxation time in a single shot experiment, and as alternative 2D pulse sequences to obtain T_1_–T_2_ and diffusion-T_1_ correlation maps. This review also presents some applications of these sequences in plant science.

## 1. Introduction

The fast and non-invasive methods to determine oil content in oilseeds were the first applications of time domain nuclear magnetic resonance (TD–NMR) in plant science, and they have since been used in germplasm evaluation and plant breeding programs [1]. The analyses of oilseeds, with low moisture content, are performed by measuring the intensity of free induction decay (FID) after a radiofrequency (rf) pulse. The measurements are performed at 50 μs to avoid interference of solid components (proteins, carbohydrates). Spin-echo sequence has been used to measure moisture and oil content. The oil content is calculated by the echo intensity at 7 ms and the moisture content by the difference between the FID (at 50 μs) and echo intensities [2].

Pulse sequences to measure longitudinal (T_1_) and transverse (T_2_) relaxation times, the self-diffusion coefficient (D), and flow [3] have been used in plant science for approximately four decades. Saturation-recovery (SR) or inversion-recovery (IR) sequences measure T_1_ [4], the Carr-Purcell-Meiboom-Gill (CPMG) sequence measures T_2_ [5], while pulse field gradient (PFG) sequences measure diffusion or flow [6,7,8,9].

Two-dimension (2D) experiments are also used in plant science to reduce peak overlap and obtain correlation between relaxation times (T_1_–T_2_) and between diffusion and T_2_ (T_2_–D) [3,10,11,12,13]. The 2D T_1_–T_2_ measurements are performed combining sequences to measure T_1_ (SR or IR), followed by the CPMG sequence, called SR–CPMG and IR–CPMG sequences, respectively. T_2_–D measurements are conducted with CPMG preceded by pulses gradient spin echo sequence (PGSG–CPMG) [11].

In the lasts two decades, one-dimensional (1D) and two-dimension (2D) pulses sequences have been proposed for TD–NMR [3,12,13,14,15]. This mini-review focuses on sequences, such as CPMG with low refocusing pulses, which minimize the sample-heating problem, and 1D and 2D sequences based on a special regime of steady state free precession, known as continuous wave free precession (CWFP).

## 2. One-Dimensional (1D) Pulse Sequences

### 2.1. CPMG with Low Refocusing Pulses

Most TD–NMR studies in plants are based on the measurement of T_2_ relaxation time obtained by the CPMG pulse sequence, which consists of a π/2 pulse in the x axis followed by time (τ) and a train of refocusing π pulses in the y axis separated by time 2τ (Figure 1 using θ = π ) [5]. The CPMG pulse sequence is very robust with negligible dependence on pulse imperfections and magnetic field inhomogeneity; in addition, a single scan provides the full relaxation decay [5,10,16].

The CPMG signals have been used to study plants in laboratories, greenhouses, and in the field [17,18,19,20,21,22]. Normally, the CPMG decay is measured for 5T_2_ to obtain suitable discrete or continuous fitting. Since T_1_ and T_2_ are similar in low field TD–NMR spectrometers, the scans can be performed without signal saturation, even with a recycle delay (RD) of a few milliseconds. This procedure is important to being able to carry out experiments in short measuring times for static samples or in moving samples on a conveyor belt. When RD is in the order of milliseconds, the π pulses are applied almost continuously, which can cause probe and power amplifier overheating [17]. In some cases, mainly for living tissues, a continuous high power rf pulse train may exceed the specific absorption rate (SAR), which can damage the sample by excessive heating and also lead to erroneous T_2_ values. The sample heating problem is a critical issue, for example, for oilseeds, as T_2_ varies exponentially with temperature [19,23]. 

To minimize instrumental or sample-heating problems, T_2_ measurements have shown accuracy using CPMG with a low refocusing flip angle (LRFA–CPMG) (Figure 1 using θ << π), which may lower rf power by up to one order of magnitude [16]. In LRFA–CPMG, T_2_ accuracy depends on experimental parameters, such as τ value and magnetic field inhomogeneity. Figure 2 shows the experimental CPMG decay using π, 3π/2, π/2, and π/4 refocusing pulses, for a soybean oil sample, using two τ values in two magnetic fields with different field homogeneity. Figure 2A,B show CPMG signals for a signal with 15 Hz line width and τ values of 0.1 (A) and 0.4 ms (B). Figure 2C,D show CPMG signals for a signal with 100 Hz line width and τ values of 0.1 ms (Figure 2C) and 0.4 ms (Figure 2D). These results show that for a quite homogeneous magnet, a very low refocusing pulse with short τ values gives a signal with the same intensity and T_2_ values as that of classical CPMG with π pulses. Signal intensity decays for longer τ values and magnetic field inhomogeneity; however, T_2_ values are still rather accurate [16]. 

Figure 3A shows CPMG decays of a castor bean seed (oil signal) obtained with the CPMG pulse sequence with π (red line) and π/2 (black line) refocusing pulses. These experiments were performed in a Minispec instrument (Bruker 20 MHz for ^1^H) using τ = 0.5 ms. Figure 3B shows the continuous T_2_ distribution of Figure 3A signals, obtained by inverse Laplace transform (ILT) [24]. These figures show that LRFA–CPMG, with π/2 refocusing pulses, shows identical T_2_ values when compared with those of the standard CPMG sequence. As B_1_∼P^1/2^, where P is the rf power and B_1_ the amplitude of the magnetic field of the applied pulse, the LRFA–CPMG with π/2 pulses reduced to one quarter of the energy deposited in the sample [16] and also obtained similar T_2_ results to those obtained by the conventional CPMG method (Figure 3).

Figure 4 shows the high correlation (*r* = 0.98) between T_2_ obtained by LRFA–CPMG performed with a π/2 refocusing flip angle with T_2_ measured by conventional CPMG for oil seeds of 30 plant species [16]. 

### 2.2. Continuous Wave Free Precession (CWFP) Sequences

#### 2.2.1. Quantitative Analyses

Continuous wave free precession (CWFP) is a special regime of the steady-state free precession (SSFP) condition [24,25,26,27,28,29,30]. The CWFP sequence consists of an equally spaced rf pulse train separated by a time interval (Tp) shorter than T_2_ [25] and shorter than T_2_* [26,27] (Figure 5)**.** Ernest and Anderson described the analytical solution for SSFP/CWFP regimes and this can be found elsewhere [28,29,30]. 

Figure 6 shows the numerical simulations of the real SSFP/CWFP signals using Block equations [31], T_1_ = 150 ms, T_2_ = 50 ms, T_2_* = 0.5 ms, and different time between pulses, Tp values, and offset frequencies. The standard SSFP regime (Figure 6A) is obtained when a train of rf pulses (gray vertical lines) with the same phase, separated by a time interval (Tp =1.45 ms), is shorter than T_2_ and longer than T_2_*, that is, T_2_ > Tp > T_2_*. In this condition, FID (blue arrows) and echo (red arrows) signals, dephased in π (180°), are observed immediately after a π/2 pulse and before the next π/2 pulse, respectively [26,27,32].

The CWFP regime is obtained when Tp is shorter than T_2_ and T_2_* (T_2_ >> Tp < T_2_*). In the CWFP regime (Figure 6B,C) FID and echo signals are overlapped and the interaction is constructive (B) or destructive (C), depending on the precession angle ψ = ϖ_0_Tp (ϖ_0_ is the frequency offset from resonance), flip angle θ, and T_1_ and T_2_ relaxation times according to Equation (1). For Tp = 0.3 ms, θ = π/2, and ψ = (2n+1)π = 8.333 kHz, the interaction is constructive, yielding a CWFP signal with maximum amplitude (Figure 6B). However, when θ = π/2 and ψ = 2nπ = 6.666 kHz, FID and echo interaction is destructive and a minimal signal is observed (Figure 6C) [28].
(1)$$\left|M_{\mathrm{SS}}\right|=\frac{M_{0}\left|sin\theta \right|\sqrt{2-2cos\psi }}{\left(1+cos\theta \right)\left(1-cos\psi \right)+\left(1-cos\theta \right)2T_{1}/T_{2}})$$

The magnitude of the CWFP signal (|M_ss_|) is constant (dashed red line on top of the CWFP signal in Figure 6B) and for ψ = (2n+1)π, it is dependent on M_0_, T_1_, and T_2_ according to Equation (2). Equation (2) also shows that (|M_ss_|) is not dependent only on T_1_, as observed in conventional NMR sequences, but on T_1_/T_2_. For example, when T_1_ = T_2_ (|M_ss_|) = 0.5M_0_.
(2)$$\left|M_{\mathrm{SS}}\right|=\frac{M_{0}}{1+T_{1}/T_{2}}$$

Therefore, in the CWFP sequence, the pulse interval can be as short as possible (Tp < T_2_*), which allows the acquisition of thousands of scans per second [26,27]. Consequently, the CWFP sequence is used to enhance the signal-to-noise ratio (SNR) for more than one order of magnitude in the TD–NMR, without increasing the experimental time [26,27]. The CWFP signal also shows linear correlation with concentration for a sample with similar T_1_/T_2_, allowing fast quantitative measurements [27]. The CWFP sequence has been used to measure oil content in seeds with low oil content, such as peas and maize, without increasing measuring time [26,33].

The CWFP sequence is also used to determine the oil content in seeds moving at 13 cm/s on a conveyor belt. In this experiment, each seed gives a signal where the intensity is proportional to the oil content. Therefore, it is a high-throughput method to determine oil content in more than 20,000 seeds per hour [33].

#### 2.2.2. CWFP Sequence to Measure T_1_ and T_2_ in a Single Experiment

The CWFP sequence can also be used to measure both relaxation times (T_1_ and T_2_) in a single shot sequence [15,26,27,32,34]. These measurements require acquiring the amplitude of the NMR signal from the first pulse to the CWFP regime, using θ = π/2 and ψ = (2n+1)π (Figure 7). After the first π/2 pulse, the signal magnitude is at a maximum and is proportional to *M*_0_. After the following pulses, the signal oscillates in a transient period depending on T_2_* (Figure 7, dark gray region). When oscillation stops, the signal reaches a quasi-stationary state (QSS), indicated by a red arrow in Figure 7. From the QSS, the signal decays to the CWFP regime with a time constant T* that depends on θ and on relaxation times, according to Equation (3).
(3)$${\;T}^{*}=\frac{2T_{1}T_{2}}{T_{1}\left(1-cos\theta \right)+T_{2}\left(1+cos\theta \right)}$$

For θ= π/2, Equation (3) becomes Equation (4) [32].
(4)$${\;T}^{*}=\frac{2T_{1}T_{2}}{T_{1}+T_{2}}$$

Equations (2) and (4) can be rearranged to Equations (5) and (6), respectively, to determine T_1_ and T_2_ using values of T*, |M_ss_|, and M_0_, of a single CWFP experiment (Figure 7).
(5)$$T_{1}=\frac{T^{*}/2}{\left|M_{\mathrm{ss}}\right|/M_{0}}$$
(6)$$T_{2}=\frac{T^{*}/2}{1-\left|M_{\mathrm{ss}}\right|/M_{0}}$$

The CWFP sequence results in signals with the highest variation in T* amplitude or the highest dynamic range (DR) for samples with T_1_ >> T_2_. On the other hand, DR is minimal when T_1_~T_2_, which makes it difficult to determine T* in low SNR signals. To overcome this situation, a second CWPF sequence was developed to measure T_1_ and T_2_ for samples with T_1_~T_2_ [35]. This sequence has a π/2 pulse separated by a Tp/2 before the CWFP pulse train (ϕ = x) (Figure 8). It is a Carr–Purcell sequence with π/2 refocusing pulses and it is named CP–CWFP [35]. Recently, the use of a π phase alternation, on the CWFP loop of pulses (ϕ = −x), has been proposed (Figure 8) [15]. This sequence, known as CP–CWFPx−x, has been the best CWFP sequence due to its highest DR for both conditions T_1_~T_2_ and T_1_ > T_2_ and the signal can also be measured on resonance (ψ = 0) [15].

#### 2.2.3. CWFP Sequence to Measure T_1_ in a Single Shot Experiment

CWFP sequences have also been used to measure T_1_ in a single shot experiment [14]. According to Equation (3), T*, of the CWFP signal, can be used to measure T_1_, when very low flip angles are used. To obtain the maximum DR, the CWFP–T_1_ sequence (Figure 9) starts with a π pulse, which inverts the magnetization, and it is followed by a train of low flip angle pulses [14].

Figure 10 shows that the single shot CWFP–T_1_ signal (θ~π/20) has a profile similar to the IR experiment (square symbols) [14]. The low SNR in the CWFP–T_1_ signal is due to the small flip angles in the pulse sequence. However, the SNR of the CWFP–T_1_ signals can be enhanced using post-acquisition digital filters, such as Savistky–Golay or wavelet filters [36]. Figure 10 shows that the CWFP–T_1_ signal has similar results to those obtained with the standard IR method; however, in a shorter measuring time [14]. 

## 3. Two-Dimensional Methods Using the CWFP–T_1_ Sequence

Two-dimension (2D) experiments are used in TD–NMR to reduce peak overlap and obtain a correlation between relaxation times (T_1_–T_2_) [37,38,39] and between diffusion and T_2_ (D–T_2_) [40,41]. Most 2D sequences use the CPMG (T_2_) to detect the direct dimension (Figure 11A) signal, which is modulated by T_1_, T_2_, or D sequences.

Recently, it has been demonstrated that a single shot T_1_ method can also be used in the direct dimension of the 2D T_1_–T_2_ TD–NMR experiment, using the CWFP–T_1_ sequence [13]. This sequence was denominated CPMG–CWFP–T_1_ and the pulse diagram is as shown in Figure 11B. The 2D-CPMG–CWFP–T_1_ maps show higher resolution in the T_1_ dimension than do methods that use CPMG as a direct dimension. Moreover, the CPMG–CWFP–T_1_ method has a shorter experimental time (up to an order of magnitude) than does IR–CPMG for the samples with T_1_~T_2_ [13].

Figure 12 shows the T_1_–T_2_ correlation maps of the signals acquired with castor bean oil at 40 °C (Figure 12A,B) and green banana at 22 °C (Figure 12C,D) inverted by a fast multi-dimensional Laplace inversion (FLI) [37,38] algorithm. IR–CPMG maps were obtained using 32 logarithmically spaced T_1_ recovery intervals, from 0.05 to 900 ms for the castor bean oil sample and from 3 to 4,500 ms for the banana sample. The CPMG data were collected using echo numbers (*n*-Figure 11A) equal to 2400 and 12,000 echoes for the castor bean oil and banana, respectively. For both samples, the τ used on the CPMG method was 0.3 ms. The T_2_–T_1_ map measured with the CPMG–CWFP–T_1_ sequence used 32 logarithmically spaced CPMG echoes from 2 to 2400 for the castor bean oil and from 2 to 12,000 for the banana sample. CWFP–T_1_ Tp was 0.3 ms, for both samples, and the number of CWFP–T_1_ loops was 1500 for castor bean oil and 7500 for banana. Using these experimental parameters, for both samples, the CPMG–CWFP–T_1_ method (Figure 12A,C) provides T_1_–T_2_ maps with higher resolution in a shorter experimental time (up to an order of magnitude) than those produced with the IR–CPMG sequence (Figure 12B,D).

Figure 12A shows two peaks of the castor bean oil, which were assigned to the two distinct pools of protons on the fatty acid chain with different mobility or inhomogeneous structural organizations [42,43]. In the CPMG–CWFP–T_1_ map (Figure 12A), the population of protons with the highest mobility (T_1_–T_2_ = 0.19–0.14 s) represents 15.2%, while the signal of the proton with the lowest mobility (T_1_–T_2_ = 0.067–0.038 s) corresponds to the greater 2D map proportion of 84.8%. The IR–CPMG map (Figure 12B) does not allow this same quantification because of its poor resolution.

In the CPMG–CWFP–T_1_ map of a banana, the strongest signal (Figure 12C: T_1_–T_2_ = 0.55–0.39 s) was assigned to water in the vacuole and corresponds to 78.1% of the signals. The intermediary intensity signal (Figure 12C: T_1_–T_2_ = 0.19–0.14 s) is attributed to water in the cytoplasm and corresponds to 19.3% of the signal. The 2D map of banana obtained by the IR–CPMG method showed no separation between water signals in the vacuole and cytoplasm. The other signals with low intensity, in IR–CPMG and CPMG–CWFP–T_1_ bananas maps, can be related to water in the cell walls or inside the starch granule [34].

The CWFP–T_1_ sequence was also used to obtain D–T_1_ correlation maps [12]. Figure 13 shows the diagram of the 2D PGSE–CWFP–T_1_ pulse sequence to obtain a D–T_1_ correlation map. The PGSE–CWFP–T_1_ method shows similar results to the conventional IR–PGSE and SR–PGSE methods; however, PGSE–CWFP–T_1_ provides a faster analysis and low SAR due to the small flip angle at the CWFP–T_1_ method [12].

Figure 14 shows the D–T_1_ correlation maps obtained by IR–PGSE (A) and PGSE–CWFP–T_1_ (B) sequences for asparagus stems oriented parallel (in black) and perpendicular (in red) to the pulse field gradient direction. Both maps show similar results; however, PGSE–CWFP–T_1_ is much faster (up to an order of magnitude) than the IR–PGSE experiment [12]. Applications of the PGSE–CWFP–T_1_ sequence to measure D–T_1_ correlation in fruits and vegetables are showing promising results. 

## 4. Conclusions

The recent 1D methods based on the CWFP regime or the low-power CPMG sequence have potential as alternatives to CPMG to enhance SNR and resolution and minimize instrumental or sample-heating problems, especially when fast analysis protocols are necessary for plant science studies. The new 2D pulse sequences (CPMG–CWFP–T_1_ and PGSE–CWFP–T_1_) based on the direct detection of CWFP–T_1_ signals are an alternative to obtain T_2_–T_1_ and D–T_1_ correlation maps, with better resolution and lower SAR than those from the standard 2D sequences and are an important advancement in the use of TD–NMR for plant science studies. 

## Figures and Tables

**Figure 1 plants-10-00833-f001:**
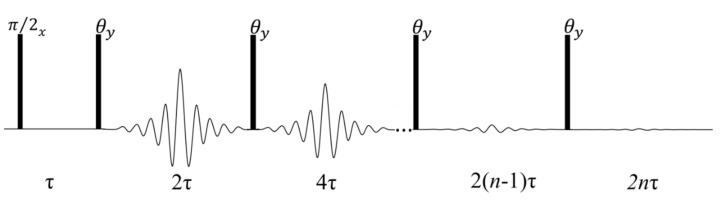
Diagram for the CPMG pulse sequence (θ = π) and for the CPMG with a low flip refocusing pulse (θ ≤ π).

**Figure 2 plants-10-00833-f002:**
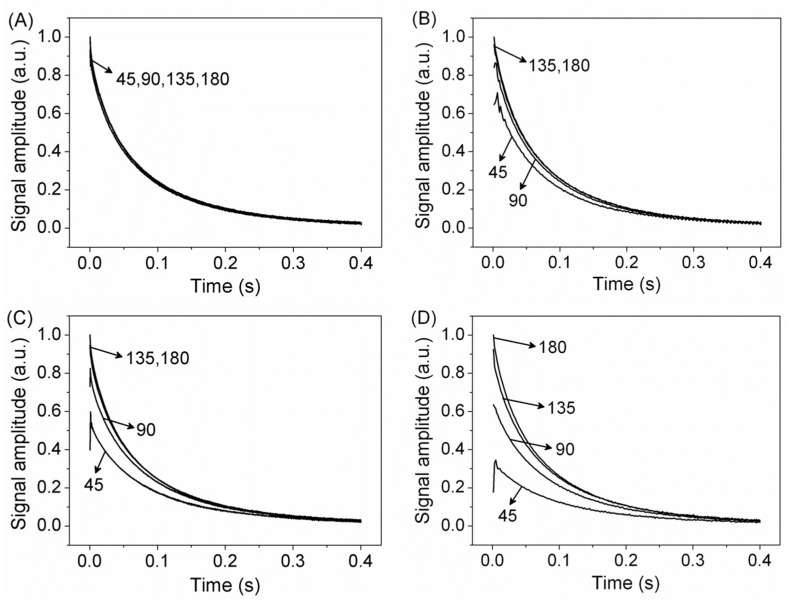
Experimental CPMG signals using refocusing angle pulse of π, 3π/4, π/2, and π/4 for soybean oil in different homogeneities (Δν = 15 Hz (**A**,**B**) and 100 Hz (**C**,**D**)), τ = 0.1 ms (left) and 0.4 ms (right). Adapted from publication [16]. Copyright (2011), with permission from Elsevier.

**Figure 3 plants-10-00833-f003:**
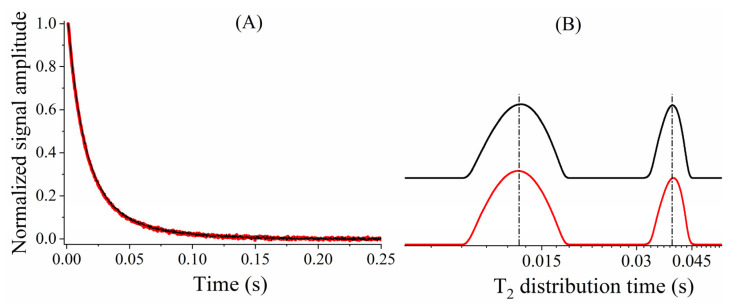
(**A**) CPMG signals and (**B**) its inverse Laplace transform (ILT) obtained from castor bean seed (oil signal) using π (red line) and π/2 (black line) as a refocusing pulse. The dashed-dotted line in (**B**) is the guide to the signal center.

**Figure 4 plants-10-00833-f004:**
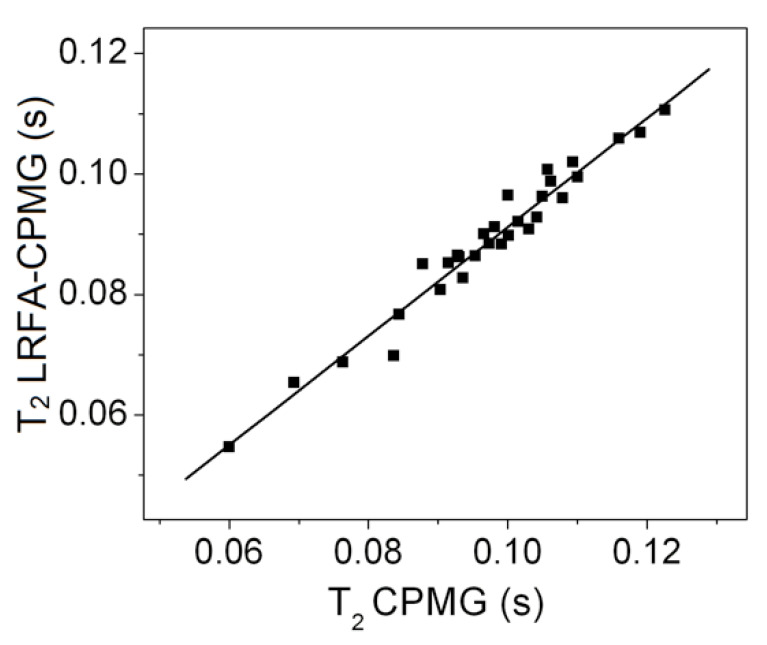
Correlation between T_2_ values obtained with the CPMG and LRFA–CPMG methods using π/2 as a refocusing and different oilseed species; *r* = 0.98. Adapted from Publication [16]. Copyright (2011) with permission from Elsevier.

**Figure 5 plants-10-00833-f005:**
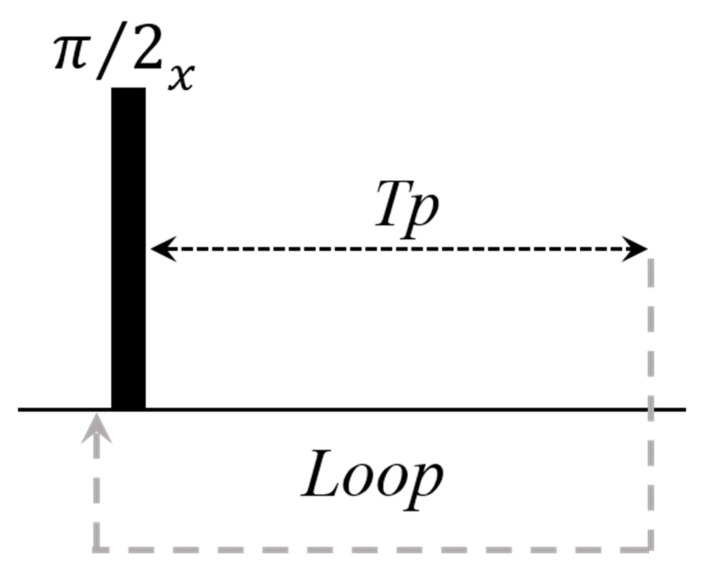
Continuous wave free precession (CWFP) pulse sequences that generate a special steady state free precession (SSFP) condition in the magnetization, when Tp < T_2_* < T_2_.

**Figure 6 plants-10-00833-f006:**
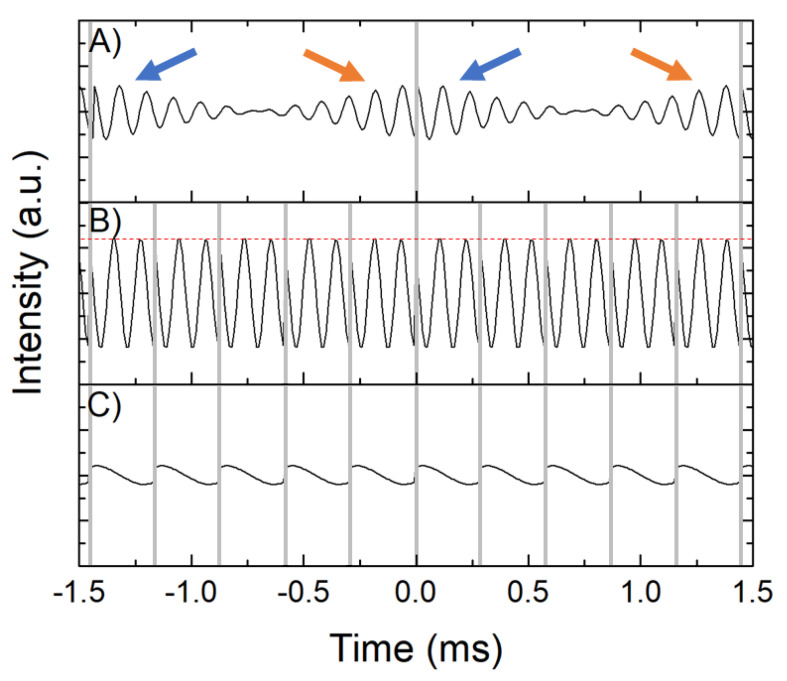
NMR signals simulated numerically using T_1_ = 150 ms, T_2_ = 50 ms, T_2_* = 0.5 ms, and different Tp values. (**A**) Tp = 2.9T_2_*, (**B**,**C**) Tp < T_2_*. The frequency offset is 8.333 kHz (**A**,**B**) and 6.666 kHz (**C**). Adapted from Publication [28].

**Figure 7 plants-10-00833-f007:**
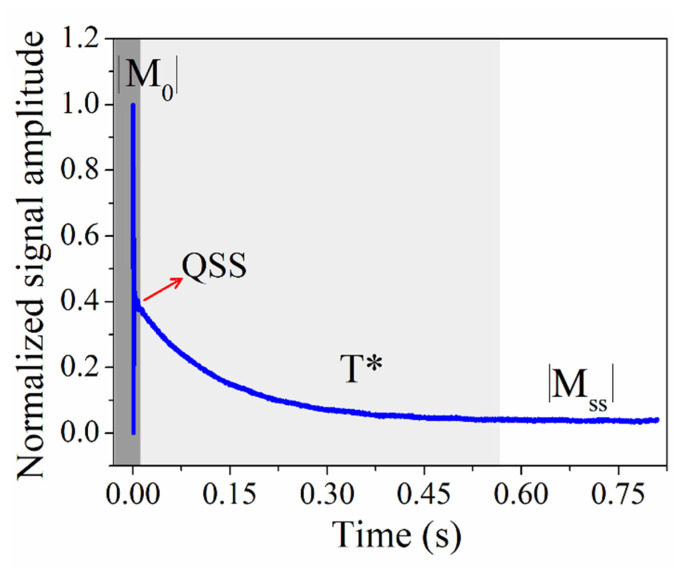
Typical CWFP signal (magnitude) obtained from the first pulse. The dark grey region indicates the initial signal that oscillates for a time that is dependent on T_2_*. When the oscillation stops, a quasi-stationary state (QSS) signal is observed (red arrow). The light grey region shows T* decay from the QSS to the steady state regime |Mss| (white region).

**Figure 8 plants-10-00833-f008:**
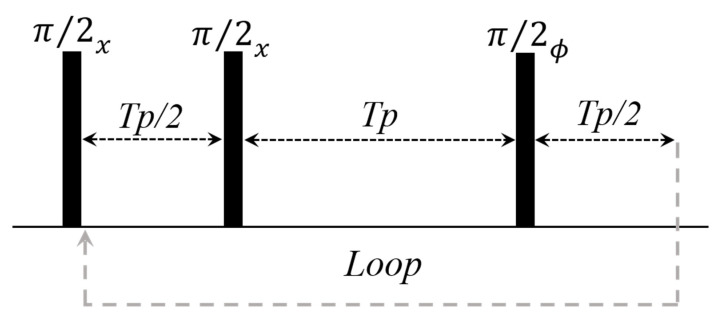
Pulse diagram for CP–CWFP (ϕ = x)/CP–CWFPx−x (ϕ = −x) pulse sequences. Adapted from publication [15]. Copyright (2015), with permission from Elsevier.

**Figure 9 plants-10-00833-f009:**
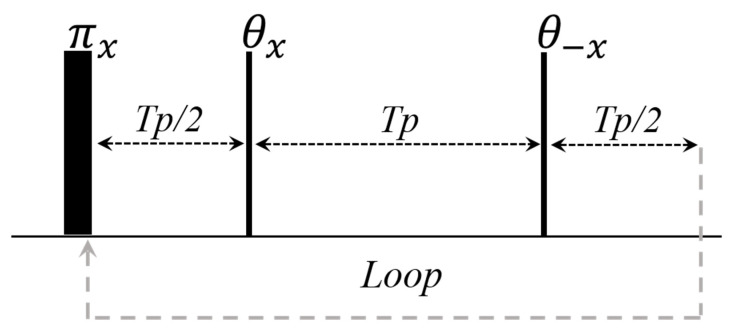
Pulse diagram for the CWFP–T_1_ pulse sequences. Adapted from publication [14]. Copyright (2016), with permission from Elsevier.

**Figure 10 plants-10-00833-f010:**
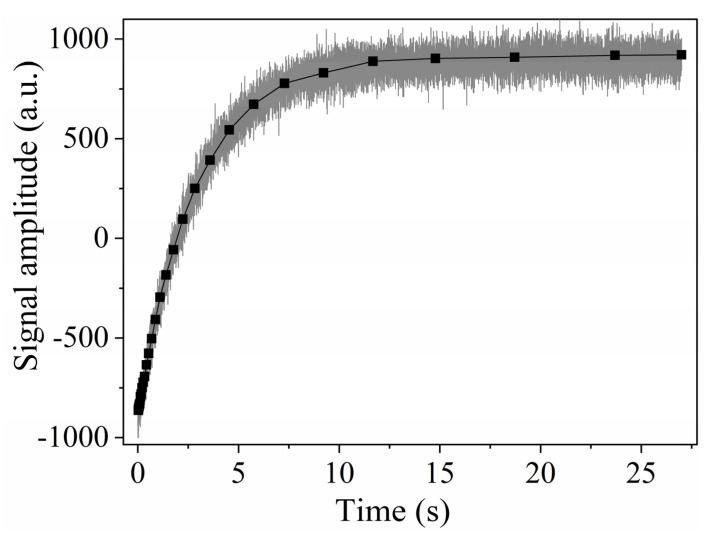
Inversion recovery signal (black points) and the CWFP–T_1_ signal (gray line) obtained for a water sample at 27 °C. Adapted from publication [14]. Copyright (2016), with permission from Elsevier.

**Figure 11 plants-10-00833-f011:**
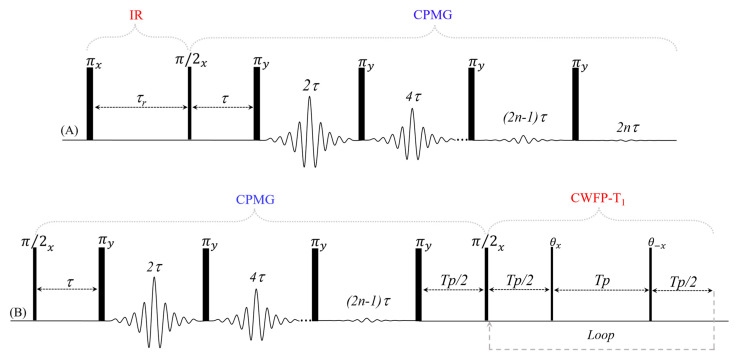
Pulse diagram for (**A**) conventional IR–CPMG and (**B**) CPMG–CWFP–T_1_ sequences.

**Figure 12 plants-10-00833-f012:**
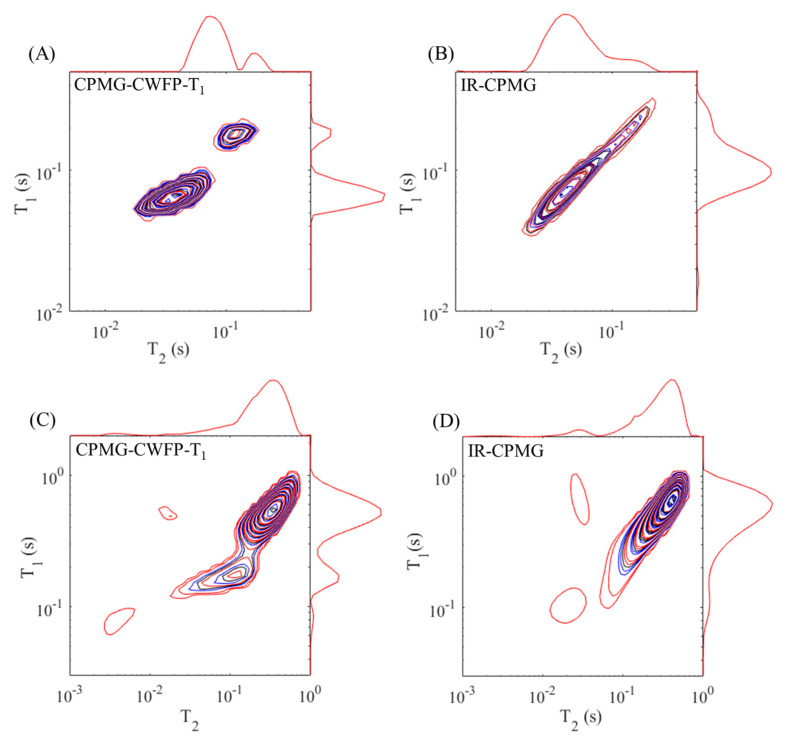
T_1_–T_2_ correlation maps for castor beans oil at 40 °C (**A**,**B**) and green banana at 22 °C (**C**,**D**) obtained by the CPMG–CWFP–T_1_ (**A**,**C**) and IR-CPMG (**B**,**D**) methods.

**Figure 13 plants-10-00833-f013:**
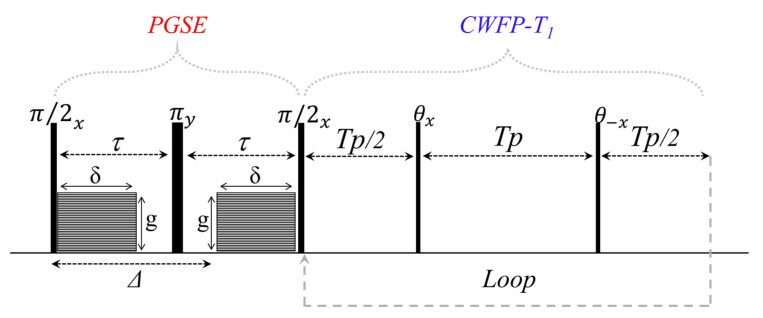
PGSE–CWFP–T_1_ diagram pulse sequence to obtain fast D–T_1_ correlation maps. Adapted from publication [12]. Copyright (2020) with permission from Elsevier.

**Figure 14 plants-10-00833-f014:**
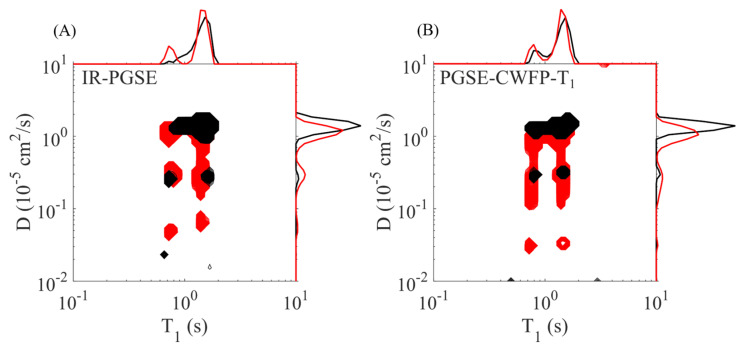
D–T_1_ correlation maps obtained by (**A**) IR–PGSE and (**B**) PGSE–CWFP–T_1_ methods for asparagus stems oriented parallel (in black) and perpendicular (in red) to the PFG. Adapted from publication [12]. Copyright (2020) with permission from Elsevier.

## Data Availability

Not applicable.

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
