# Peer review of "Recent 1D and 2D TD–NMR Pulse Sequences for Plant Science"

_plants, 2021, doi:10.3390/plants10050833_

Round 1
Reviewer 1 Report
I enjoyed reading this manuscript. It is appropriate for this special issue. My only comment is to make sure that as many english language issues are caught prior to publication.
Author Response
I enjoyed reading this manuscript. It is appropriate for this special issue. My only comment is to make sure that as many english language issues are caught prior to publication.
Answer. The new manuscript was revised by an English proof-reading service and we think that the quality of revised manuscript is improved, all changes are highlighted in the manuscript. We are grateful to the reviewer for their evaluation of our work.
Reviewer 2 Report
This review focus on TD-NMR pulse sequences for 1D and 2D relaxometry and diffusometry and their (potential) application in plants science. Emphasis is put on those using lower refocusing flip angle, i.e., Continuous Wave Free Precession (CWFP), as their intrinsic temporal resolution and low SAR make them suited for online quality control and safe in regard to energy deposition on sample. The paper is well organized and pulse sequences along with acquisition and processing strategies are well summarized. Appropriate references dealing with their application in quantitative NMR analysis of plant-based foods (mainly oilseeds) are also provided. As such the review falls well in the scoop of this special issue, even if authors already published similar reviews, the present one will be of relevance to the community of Plants readers looking for new TD-NMR approaches of its kinds for their researches.
Before acceptance for publication, authors are invited to look at minor changes suggested in form of comments in the pdf file.

Author Response
This review focus on TD-NMR pulse sequences for 1D and 2D relaxometry and diffusometry and their (potential) application in plants science. Emphasis is put on those using lower refocusing flip angle, i.e., Continuous Wave Free Precession (CWFP), as their intrinsic temporal resolution and low SAR make them suited for online quality control and safe in regard to energy deposition on sample. The paper is well organized and pulse sequences along with acquisition and processing strategies are well summarized. Appropriate references dealing with their application in quantitative NMR analysis of plant-based foods (mainly oilseeds) are also provided. As such the review falls well in the scoop of this special issue, even if authors already published similar reviews, the present one will be of relevance to the community of Plants readers looking for new TD-NMR approaches of its kinds for their researches.
Before acceptance for publication, authors are invited to look at minor changes suggested in form of comments in the pdf file.
Answer.
Pag. 1, Line 13 and 24. We agree that “plant science” is an appropriate term and it was used in the manuscript.
Pag. 1, Line 26-31. Corrected. The text was modified from line 23 to 36.
Pag. 4, Line 114. Corrected. “shorter than”.
Pag. 5, Line 127. We agree that T2 is never shorter than T2* and it is redundant to write (T2 > T2*), so we modified the text to: “ (T2 >> Tp < T2*) ”
Pag. 6, Line 174. Corrected. “in this condition is more difficult to determine T* in low SNR signals.”
Pag. 7, Line 196. Corrected. “shows”
Pag. 7, Line 200. Corrected. “shows that”
Pag. 7, Line 202. Corrected.
We are very grateful to the reviewer for their careful evaluation of our work and we think that the quality of revised manuscript is improved thanking in account the reviewer’s suggestions.
Reviewer 3 Report
This mini-review is a nice and well-structured overview of previous work by the authors.
The main points of attention/criticism:
1) The English/language would need some attention.
2) The level of explanation/readability is unbalanced:
The introduction and section 2.1 are well written and understandable for a wider audience interested in this topic, even lacking the technical background.
Section 2.2.1, in sharp contrast, is very technical and even puzzling to somebody with a background in the field. Instead of trying to be complete on the topic, readers would be more helped by a simplified explanation of the conceptual differences in the approaches presented. The focus on figure 6 is not helpful given the (expected) audience.
Sections 2.2.2 and further are again sufficiently approachable to appreciate the advantages the alternative approaches described.
The final part of section 3 on D-T1, feels mainly as an technical achievement without much attention for possible applications.
3) The tone of the paper is, especially in the second half, sometimes not very scientifically; one could call it even subjective:
Attention should be paid to unsubstantiated and at unspecified qualifications like: “a shorter experimental time”, “higher resolution”, and “is much faster than”, to name some examples.
Author Response
This mini-review is a nice and well-structured overview of previous work by the authors.
The main points of attention/criticism:
1) The English/language would need some attention.
Answer. The new manuscript was revised by an English proof-reading service and we think that the quality of revised manuscript is improved, all changes are highlighted in the manuscript. We are grateful to the reviewer for their evaluation of our work.
2) The level of explanation/readability is unbalanced: The introduction and section 2.1 are well written and understandable for a wider audience interested in this topic, even lacking the technical background. Section 2.2.1, in sharp contrast, is very technical and even puzzling to somebody with a background in the field. Instead of trying to be complete on the topic, readers would be more helped by a simplified explanation of the conceptual differences in the approaches presented. The focus on figure 6 is not helpful given the (expected) audience.
Answer. In Section 2.2.1 we introduced minimal information about the Continuous Wave Free Precession (CWFP) pulse sequences that we think that is necessary for the journal audience. We think that Figure 6 is helpful because it shows how the reduction of the time between pulses (Tp) the magnetization goes from the steady state (SSFP) to Continuous Wave (CWFP) regime. We are grateful for the reviewer suggestion to simplify the text for a broad audience, but usually this demonstration of the steady state formation is the most intuitive way to introduce the technique.
Sections 2.2.2 and further are again sufficiently approachable to appreciate the advantages the alternative approaches described. The final part of section 3 on D-T1, feels mainly as an technical achievement without much attention for possible applications.
Answer. Section 2.2.2 describe the main application of CWFP in the determination of T1 and T2 relaxation times in one experiment. The mathematical description is necessary for the correct evaluation of the parameters and how to extract the information from the signals. The final part of section 3 on D-T1, we add a text “Applications of the PGSE-CWFP-T1 sequence to measure D-T1 correlation in fruits and vegetables are showing promising results.”
3) The tone of the paper is, especially in the second half, sometimes not very scientifically; one could call it even subjective: Attention should be paid to unsubstantiated and at unspecified qualifications like: “a shorter experimental time”, “higher resolution”, and “is much faster than”, to name some examples.
Answer. These sentences occurred because the experimental time is very dependent of the kind of samples that are in evaluation, such as rigid materials like wood and polymers, or more viscous as oil, or less viscous water-based materials, all have different orders of magnitude in the relaxation times, and so in the total experimental time. We agree with the reviewer that a more objective sentences are necessary, and a range of values were included in some sentences in the text. Pag. 8, line 212, Pag. 9, line 228, Pag. 10, lines 259.
Reviewer 4 Report
The manuscript is a mini review submitted to Plants. It summarizes 1D
and 2D pulse sequences designed with low field (time domain) NMR in
mind. Its applications to plant materials are demonstrated. The review
is focused on the developments of low flip angle and continuous wave
free precession pulse sequences for rapid acquisition and reduced rf
pulse induced heat deposition. Most of the review summarises the
contributions of the corresponding author over the last decade. This
is fine if it is the intention of the review to do so and if this is
in line with the journal guidelines (I do not know if this is a review
for a focus or special issue).
The manuscript is mostly well written with some minor issues outlined
below. I recommend further proofreading by a native English speaker.
I suggest publication after the authors have considered the comments
below.
- "The first applications of time domain nuclear magnetic resonance
(TD-NMR) in plant science were the fast and non-invasive methods for
determination of oil content in oilseeds, which has been used in
germplasm evaluation and plant breeding programs [1]."
The reference used here is a review from 2017. This may create the
impression that first applications for TDNMR in plant science are
only 5 years old if a rushed reader mistakes the review [1] for an
original research paper. Moreover, the review cited as ref [1] then
points further back to publications from 1993, 1995 and 1999. That
means as a reader I need to go back to the original literature via
the review from 2017 instead of having the source referenced
directly in the publication I am reviewing here. I therefore suggest
to cite original papers directly (besides the review paper from
2017) or clarify that the method is used at least since 1993 and
state that more details and further publications are provided in [1].
- "Since T1~T2 in low field TD-NMR spectrometer, the scans can be
performed without signal saturation even with minimal recycle delay
(RD) time." I am not sure what the authors mean here. What does
"signal saturation" refer to? Do the author mean that T1 is so
short that the 5xT1 can be achieved even for short repetition times
(which is what the authors call RD, I guess)? And what does
"minimal" mean? An optimum regarding NMR properties or a limit
coming from the NMR apparatus?
- It would be helpful to know how the line width has been controlled
in [15]. Were the experiments carried out on different magnet
systems or has a magnet been "de-shimed"?
- "... a standard commercial instrument... " I am not aware of a
standard for commercial instruments. Please provide details of the
NMR system used.
- "... and echoes (red arrows) signals, dephased in π, are observed
immediately after... " What does "dephased in π" mean?
- "... Equations 2 and 4 can be rearranged to equations 5 and 6 to
measured T 1 and T 2 using measuring... " should perhaps read
"... Equations 2 and 4 can be rearranged to equations 5 and 6 to
determine T 1 and T 2 using measuring... "
- "... On the other hand, the DR is minimal when T 1 ~T 2 , that
difficult the determination ..." should perhaps read "... On the
other hand, the DR is minimal when T 1 ~T 2 , which makes it
difficult to determine ..."
- It seems that the data in Fig. 12 are unpublished data obtained with
the method published in [12]. I take some issue with the statement:
"...IR-CPMG map (Figure 12B) does not allow this same quantification
because of its poor resolution." The IR-CPMG map could have been
acquired with a large number of delay times in the IR dimension for
the sake of higher resolution and comparability with the CWFP
sequence. There is no way of telling what the ground truth is unless
the authors would provide more evidence that the continuous
distributions in Fig 12 b) and d) are inaccurate.
Author Response
The manuscript is a mini review submitted to Plants. It summarizes 1D and 2D pulse sequences designed with low field (time domain) NMR in mind. Its applications to plant materials are demonstrated. The review is focused on the developments of low flip angle and continuous wave free precession pulse sequences for rapid acquisition and reduced rf pulse induced heat deposition. Most of the review summarises the contributions of the corresponding author over the last decade. This is fine if it is the intention of the review to do so and if this is in line with the journal guidelines (I do not know if this is a review for a focus or special issue).
The manuscript is mostly well written with some minor issues outlined below. I recommend further proofreading by a native English speaker.
The new manuscript was revised by an English proof-reading service and we think that the quality of revised manuscript is improved, all changes are highlighted in the manuscript. We are grateful to the reviewer for their evaluation of our work.
I suggest publication after the authors have considered the comments below.
- "The first applications of time domain nuclear magnetic resonance (TD-NMR) in plant science were the fast and non-invasive methods for determination of oil content in oilseeds, which has been used in germplasm evaluation and plant breeding programs [1]."
The reference used here is a review from 2017. This may create the impression that first applications for TDNMR in plant science are only 5 years old if a rushed reader mistakes the review [1] for an original research paper. Moreover, the review cited as ref [1] then points further back to publications from 1993, 1995 and 1999. That means as a reader I need to go back to the original literature via the review from 2017 instead of having the source referenced directly in the publication I am reviewing here. I therefore suggest to cite original papers directly (besides the review paper from 2017) or clarify that the method is used at least since 1993 and state that more details and further publications are provided in [1].
Answer. We agree with the reviewer. An original reference for TDNMR in plant science was adding to the paper: new Reference [1], Pag. 1 Line 25.
- "Since T1~T2 in low field TD-NMR spectrometer, the scans can be performed without signal saturation even with minimal recycle delay (RD) time." I am not sure what the authors mean here. What does "signal saturation" refer to? Do the author mean that T1 is so short that the 5xT1 can be achieved even for short repetition times (which is what the authors call RD, I guess)? And what does "minimal" mean? An optimum regarding NMR properties or a limit coming from the NMR apparatus?
Answer. The sentence was rewritten (page 2) to clarify the questions raised by the reviewer
Since T1 and T2 are similar in low field TD-NMR spectrometers, the scans can be performed without signal saturation even with recycle delay (RD) of few miliseconds.
- It would be helpful to know how the line width has been controlled in [15]. Were the experiments carried out on different magnet systems or has a magnet been "de-shimed"?
Answer. The experiments were performed in different magnets (permanent magnets), which have a specific width line standard of the commercial equipment. No modifications were made in the shimming. Lines 75-76 was change for better comprehension: “… two magnetic fields with different field homogeneity.”
- "... a standard commercial instrument... "I am not aware of a standard for commercial instruments. Please provide details of the NMR system used.
Answer. The sentence “…a standard commercial instrument…” was change to “…a Minispec instrument (Bruker 20 MHz for 1H) …”. Pag. 3, Line 90.
- "... and echoes (red arrows) signals, dephased in π, are observed immediately after... " What does "dephased in π" mean?
Answer. Means that the FID and echo are 180° out of phase. Sentence “(180°)” included in Pag. 5, Line 125.
- "... Equations 2 and 4 can be rearranged to equations 5 and 6 to measured T 1 and T 2 using measuring... " should perhaps read "... Equations 2 and 4 can be rearranged to equations 5 and 6 to determine T 1 and T 2 using measuring... "
Answer: The sentence was rewritten as suggested by the reviewer: Pag. 6, Line 165-166.
- "... On the other hand, the DR is minimal when T 1 ~T 2, that difficult the determination ..." should perhaps read "... On the other hand, the DR is minimal when T 1 ~T 2, which makes it difficult to determine ..."
Answer: This sentence was rewritten as suggested: Lines 173-174: … On the other hand, the DR is minimal when T1~T2, which makes it difficult to determine T* in low SNR signals”
- It seems that the data in Fig. 12 are unpublished data obtained with the method published in [12]. I take some issue with the statement: "...IR-CPMG map (Figure 12B) does not allow this same quantification because of its poor resolution." The IR-CPMG map could have been acquired with a large number of delay times in the IR dimension for the sake of higher resolution and comparability with the CWFP sequence. There is no way of telling what the ground truth is unless the authors would provide more evidence that the continuous distributions in Fig 12 b) and d) are inaccurate.
Answer. The authors agree with the reviewer. More information about the experimental acquisitions parameters was included in the article: Pag. 8 and 9, lines 218-230. For those experimental condition, the CPMG-CWFP-T1 had more resolution than IR-CPMG method.